# Facilitating Community Risk Communication for Wide-Area Evacuation during Large-Scale Floods

**DOI:** 10.3390/ijerph16142466

**Published:** 2019-07-11

**Authors:** Takeyasu Suzuki, Takanori Watanabe, Shin’ichiro Okuyama

**Affiliations:** 1Disaster and Environmentally Sustainable Administration Research Center, University of Yamanashi, Kofu City, Yamanashi 400-8511, Japan; 2Sanpoh Co., Ltd., Kai City, Yamanashi 400-0111, Japan; 3Civil and Environmental Engineering Course, Integrated Graduate School of Medicine, Engineering and Agricultural Sciences, University of Yamanashi, Kofu City, Yamanashi 400-8511, Japan

**Keywords:** risk communications, CAUSE model, BECAUSE model, wide-area evacuation, expected maximum rainfall, community disaster management plan, flood, role of stakeholders

## Abstract

Large-scale floods have been occurring more frequently in Japan as a result of current global weather anomalies, yet evacuation procedures face several issues. These include low evacuation rates of citizens, wide-area evacuation by car, and residents who cannot evacuate on their own. For example, in the Kofu Basin, Yamanashi Prefecture, due to the size of the potential inundation area and a population that exceeds 300,000 people spread across 10 municipalities, a large number of residents would have to evacuate across municipal boundaries by car. The author proposed and applied a risk communication method to the Riverside District, Chuo City (with about 1400 households and a population of about 4000), assisting in developing a community disaster management plan for wide-area evacuation without a single victim in case of floods, which has been in place for three years. The next step was risk communication to key stakeholders, such as national, prefectural, and municipal governments. Finally, a public symposium on large-scale evacuation in the Kofu Basin was held. During the panel discussion with representatives of the Kofu River and National Road Office, prefectural government of Yamanashi, the municipality, community residents, and the author as panelists, the role of each stakeholder in area-wide evacuation was clarified and confirmed.

## 1. Introduction

As an archipelagic region, Japan is prone to natural disasters. Firstly, approximately 20% of globally damaging earthquake epicenters of 6.0 or greater on the Richter scale and approximately 7% of active volcanoes in the world are distributed within and around the Japanese archipelago, mainly due to the collision of four plates and complex plate tectonics. Secondly, Japanese mountains are steep because of active orogenic movement. Thirdly, Japan is annually plagued by typhoons because of its geographical location within the Asian monsoon zone [1]. Japan has been exerting ample effort in reducing the damage caused by these frequent disasters through non-structural measures (such as a legal system) and structural measures, to the extent that it has established a leading position in the world in terms of disaster preparedness and formulating countermeasures.

On a parallel note, the number of typhoons frequenting Japan, including the amount of precipitation, has been increasing in recent years because of the global weather anomaly. Moreover, whereas both the national and local governments have strengthened and implemented strict laws on disaster mitigation, preparedness, and response, people’s awareness of self-help and mutual help during a disaster has weakened, thereby raising the tendency to rely on administrative aid for disaster management [2]. Therefore, despite the imminent dangers such as river flooding, sediment disasters, and tsunamis, it is a serious problem that residents do not evict and evacuate.

This study discusses the impact of a large-scale flood scenario. In 1959, approximately 5098 human casualties were recorded on the landing of the Isewan Typhoon, the largest in Japan’s recorded history of typhoon damage. This resulted in the introduction of the Disaster Countermeasures Basic Act; since the enforcement of this Act, the number of victims of flood damage has never exceeded 1000 per year in Japan. Furthermore, as a result of revisions to the River Act and the Flood Control Act, along with the Disaster Countermeasures Basic Act and the implementation of flood control projects, the number of human casualties due to the storm and flood damage from 1994 to 2013 was 1192, meaning roughly 50 victims a year [1]. However, because of heavy rains associated with global warming, the scale of the damage caused by flooding has increased, and floods of mountain rivers, which are a combination of debris flow and flooding, are also more common [3].

Disasters occur when an incident exceeds nature and society’s ability to manage it. The increase in heavy rainfalls tends to overwhelm both structural and non-structural measures, and there is concern that this damage will only worsen in the future. Structural measures are indispensable, but it takes much time for them to be established. The measures that can be taken at an early stage are non-structural, and among them, raising people’s awareness of disaster management has to be improved. Thus, the realization of early eviction and evacuation to a safe place has become an urgent issue.

## 2. Current States of Wide-Area Evacuation and Objectives of this Study 

The 2015 Kanto and Tohoku heavy rains resulted in massive flooding of the Kinugawa River and 40 km^2^ or one-third of the area of Joso City in Ibaraki Prefecture, leaving two human casualties. At the height of the flood, more than 4000 people were stranded in the flooded area and later rescued by helicopters and boats [4]. The survey conducted by Irie [4] indicated that the largest number of evacuees, accounting for 35% of the total, was headed outside Joso City via car transport (89% of all related evacuation means). In terms of motive, 41% of the evacuees were “family and friends prompted.” Moreover, most of those affected indicated that their reason to remain indoors was due to the prejudice of normality, believing that the heavy rain would not inundate, or that it was difficult to move “considering the elderly.” The reason “neighbors are not evacuating” rated high as well, accounting for 12% of the survey responses [5].

Accordingly, the July 2018 western Japan heavy rain disaster caused the Oda River, Suematsu River, and Takama River to overflow and floods flowed into Mabi-cho, Kurashiki City, Okayama Prefecture. Approximately 27% or 27 km^2^ of Mabi-cho was flooded leaving 51 human casualties, of which 45 (or 88%) were elderly (over 65 years of age) [6]. At least 42 people out of 51 victims were persons who needed help for evacuation [6]. The evacuation rate during floods in Japan in recent years has not reached 10%, and in particular, many of the victims were persons who could not evacuate on their own.

Typically, such large-scale flooding causes houses to collapse and other properties or structures to wash away from break points of rivers. On such a note, evacuation should be judged not only by the depth of inundation but also by the presence of a dike nearby, where banks may break and intensify water flow. Figure 1a shows the damage near the break point of Joso City during the 2015 Kanto Tohoku heavy rainfall. Similarly, Figure 1b illustrates the extent of damage caused by the 2018 Western Japan heavy rainfall in Mabi-cho. 

As mentioned earlier, a wide-area scheme is necessary to facilitate early evacuation across municipal boundaries before an incoming disaster, based on the lessons learned from flooding in Joso City and Mabi-cho. Such a scheme should prioritize support to persons who need help during evacuation which would require transportation by the municipalities to welfare facilities, along with evacuation support by the neighborhood [7]. Hence, residents and the administration must work together to support the evacuation of those who cannot evacuate without assistance. To achieve wide-area evacuation of a community, it is important for residents to have the intention of evading and evacuating; a system is created that encourages them to do so, with this encouragement coming from not only their families and friends but also their neighbors. The role of the administration in wide-area evacuation is traffic control, guidance that enables evacuation, and securing of vehicle evacuation routes. Particularly important is securing public shelters, including welfare facilities, and prior coordination among municipalities is therefore indispensable [7].

The author developed a risk communication method mainly characterized by the integration of two elements: (a) CAUSE, for training on the creation of a disaster prevention system for local governments and community residents [8], and (b) BECAUSE, for cooperation among disaster-management-related organizations [9]. This study also conducted demonstrative research. The focal objectives herein were basically aligned with (i) introducing efforts on risk communication implemented in recent years for the wide-area evacuation of people from the Kofu Basin in case of the largest flood expected and (ii) developing schemes to strengthen the resilience of communities in the Kofu Basin during floods through a cooperative partnership between community residents and regional administrations.

## 3. Flood Risk in the Kofu Basin

Residents who do not evacuate even when normal evacuation information is issued will impede the success of a well-organized wide-area evacuation scheme. A wide-area evacuation plan is defined as a strategy that spans several municipalities and is organized by prefectures. However, scheme proposals from both prefectural and municipal governments will not be readily accepted by community residents, unless residents are convinced of the necessity to evacuate. As such, a more efficient community disaster management plan on wide-area evacuation could be formulated by considering the proactive activities of the residents, so that the intent of municipality support as well as the coordination scheme between municipalities, prefectures, and residents can be clarified. This study asserts that an effective formula should be created in a bottom-up structure, where community residents play an active role in establishing evacuation plans.

Figure 2 shows the location of the Kofu Basin in Yamanashi Prefecture. The Eurasian Plate and the North American Plate collide in the east and west, respectively, and the highest mountain range in Japan is formed to the west of the basin. Meanwhile, the Philippine Sea Plate collides with the two plates from the south, and subducts under the Eurasian Plate and the North American Plate. The Kofu Basin is located at the collision center of the three plates and has a mysterious structure with an inverted triangle. Additionally, it is located in the Mount Fuji volcanic belt, with Mount Fuji to its south. Aesthetically, it is surrounded by beautiful mountain greenery and hot springs. However, the collapsed earth located in the north of the basin due to volcanic activities is deeply deposited in the subsurface of the basin. On the basin perimeter, fans are formed by the supply of debris flow deposits from mountainous rivers. During heavy rainfall, sediment and water from the surrounding mountains gather in the basin. Particularly because rivers drained flood water to the center of the basin, crops in the land could not be harvested. Since the Middle Ages, dikes have been purposely constructed and flood control schemes have been implemented to move the river channel to the edge of the basin, in an effort to turn the land into a suitable area for crop cultivation.

Figure 3 shows the main rivers and possible inundation on the Kofu Basin. The inundation area, due to the expected maximum rainfall (a probability of 1 every 1000 years), is demonstrated on the hazard map according to the categorization of the Ministry of Land, Infrastructure, Transport, and Tourism (MLIT), and the inundation area of Mabi-cho, Kurashiki City, Okayama is also shown on the same scale. In comparison, Kofu Basin has a larger inundation area (specifically, more than 10 times larger than that of Mabi-cho), which indicates that the evacuation distance exceeds 10 km, depending on the location in the basin; therefore, evacuation by car is more suitable. For example, out of a total population of 22,000 people, 51 died during the July 2018 heavy rainfall in Mabi-cho, representing a mortality rate of 0.23%. 

The mortality rate of 0.23% exceeds the mortality rate of 0.22% for the Isewan typhoon, which is the highest level in Japan’s history. Therefore, the number of victims can be estimated to be 690 people if it is calculated by multiplying the population of approximately 300,000 in the inundation assumed area by this mortality rate as a possible impact of flood damage in the Kofu Basin. However, given the low evacuation rate performance of Yamanashi Prefecture in recent years, this figure is underrepresented.

## 4. Risk Communication Models for Wide-Area Evacuation

### 4.1. CAUSE Model

The CAUSE model was proposed by Katherine Rowan as a risk communication method aimed at educating regional crisis managers [10,11]. The acronym CAUSE literally represents Confidence, Awareness, Understanding, Satisfaction with the proposed solution, and Enactment, which are also the essential stages of risk communication. Here, the crisis manager is assumed to be in an administrative job, such as a police officer or a fireman, whose communication counterpart is the general population. That is, crisis managers need to learn what people do at each stage of the CAUSE model to provide them with risk awareness, understand their content, accept the solutions presented by the risk managers, and put them into action.

Although the CAUSE model proposed by Rowan is an educational process for local crisis managers to communicate with the general population, the CAUSE model developed in this study is a risk communication method that improves the disaster response ability of the municipalities and the community resilience of the residents. Here, S (together with Satisfaction) includes the stage of the solution, where participants propose solutions or decide on solutions by themselves.

### 4.2. Community Disaster Management Plan

The Riverside District in Figure 3 is the name of a community indicating the so-called Riverside Town in Chuo City, Yamanashi Prefecture. It is located on the left bank of the Kamanashi River and was once a water reservoir sandwiched between a dike and the left bank of Kamanashi River. The Usuinuma swamp was formed after the flood of the Kamanashi River in 1907. Reclamation of the swamp began in 1959, and residential area construction has been carried out as the largest residential district (a future population of 7000) in the Kofu Basin. Nearly 1400 households and 4000 residents live in the area. Residential land started being sold from the north area, represented by three residents’ associations: the 1st Association of the North, the 2nd Association of the Central, and the 3rd Association of the South. Only in the 3rd district is residential land sales still being carried out even now.

Table 1 shows the demographics of the Riverside District. However, the figures in the table are the population of residents belonging to the residents’ associations, and the populations not belonging to the associations are not included. The bracketed numbers in 2019 are estimated population, regardless of whether they belong to a residents’ association or not.

In 2015, the author received a request from the 3rd Residential Association (about 700 households) for risk communication (Communication), which has since been initiated to support the formulation of a wide-area evacuation plan with a single casualty. The negotiations conducted between the author and the residents are described in Reference [12]. During the initial phase, a questionnaire survey on flood risk in the district was conducted (with an estimated recovery rate of 65%) targeting approximately 1400 households in the entire Riverside District. Through this questionnaire, community residents were made aware of the commonness of damage caused by floods in this district and Joso City (Awareness). Next, the author requested all households in the district to circulate the survey report, which summarizes the questionnaire results in an easy-to-understand manner. Residents shared the flood risk of the district and understood the necessity of an evacuation support system, especially for those needing help during evacuation (Understanding).

The author proposed the establishment of a system in which neighbors support those who need help during evacuation. Alternatively, district residents suggested that one should hang a towel on the second floor as a sign to inform completion of evacuation. It was decided to implement these suggestions (Satisfaction and Solution). Therefore, a wide-area evacuation plan, in which all the district residents began evacuation outside the district after evacuating those needing help, was formulated by the 3rd Residents’ Association in March 2018 (Enactment) [12]. Subsequently, representatives of the 1st and 2nd Residents’ Associations participated in this risk communication, learned, and joined the activities of the 3rd Residents’ Association in the succeeding year.

Japan has established the Disaster Countermeasure Basic Act as a comprehensive and long-term plan at the national level and a regional disaster management plan for prefectural and municipal governments at the local level, and has implemented disaster management activities at each level. 

However, in the 2011 Great East Japan earthquake, it was strongly recognized that disaster countermeasures after a large-scale wide-area disaster work well with the mutual cooperation of self-help, mutual help, and public help. On the basis of the lessons learned, some provisions concerning self-help and mutual help were added in the Disaster Countermeasure Basic Act in 2013. At that time, from the point of view of the promotion of disaster management activity by mutual help in the local community, a community disaster management plan system regarding voluntary disaster management activities conducted by residents and businesses (community residents, etc.) of certain districts in municipalities was formulated.

In 2018, another process of the CAUSE model was applied to support the community disaster management plan of the entire Riverside District (from the 1st through the 3rd Residents’ Associations). At that time, the evacuation simulation by cars was presented to representatives of the residents’ associations during the “Awareness” stage. At the stage of “Understanding”, a questionnaire survey was conducted for all households in the district (using explanatory materials through screenshots of evacuation simulation by cars shown in Figure 4) to enable residents to understand the necessity of evacuation by cars, with a time difference in each block divided in the district. Thus, in the Riverside District in Chuo City, Yamanashi Prefecture, the community disaster management plan for wide-area evacuation was formulated in the entire district in May 2019 (1400 households of the 1st through 3rd residents’ associations with 4000 people).

Figure 5 is a schematic representation of the wide-area evacuation plan for the Riverside District. Figure 6 shows the meeting conducted to build a support system for those who need help in evacuation. Figure 7 is the “My Timeline” of evacuation behavior (and preparation) distributed to each household. When the authors presented such a timeline in the form of dividing the evacuation behavior shown in Figure 5 into 10 stages, the residents added some ideas to finally create the timeline shown in Figure 5. The timelines were printed on B5 size glossy paper in black and red two-color printing and were distributed to all the households. According to the tabulation results, the recovery rate of collection slips was 65%, where the timeline had already been created at 80%. Thus, more than 52% of all households in the district were involved in the creation of the timelines.

In Japan, there is a wealth of disaster information, such as weather warnings and alerts, special alerts, recorded short-term heavy rainfall information, flood forecasts, etc. This disaster information is directly delivered to residents through various forms of media, such as the internet, television, and radio. The Ministry of Land, Infrastructure, Transport and Tourism (MLIT) recommends that residents voluntarily evacuate at an early stage by creating a timeline for evacuation based on disaster information. However, it is not easy for the general population to acquire the level of expertise for judging a situation, based on weather and river basin information. Because the river administrator (river expert) can predict the occurrence of large-scale flooding and transmit this to the mayor of the municipality, Chuo City is often able to issue evacuation preparation information earlier than usual (one day before). With this information as a trigger, the authors have proposed a plan to start wide-area evacuation action in the Riverside District. As a means of communication with district residents during disasters, LINE is used, which is one of the social networking services.

## 5. Role of Stakeholders in Wide-Area Evacuation

### 5.1. BECAUSE Model

The author developed the BECAUSE model as a training process that fosters the ability of local government practitioners to communicate with organizations inside and outside the agency for the facilitation of disaster response. Therefore, the meaning of each step of C · A · U · S · E in BECAUSE was somewhat different from the CAUSE model. In addition, BE refers to the stage of preparation for training. It is necessary to build an environment where practitioners can take part in the training, and BE is a stage to gain familiarity with the governor, mayor, and senior staff of the local administration. Moreover, S in BECAUSE means “Satisfaction”, which includes the stage of the local government practitioners proposing a “Solution” on their own or a “Solution” that shows a new method in disaster response [9]. In 2013, the BECAUSE model was applied to the construction of a regional cooperation system for wide-area evacuation, having been classified into three groups: a disaster-affected municipality; neighboring municipalities supporting it; and a supporting organization made up of disaster management and construction departments of prefectural government, prefectural police headquarters, and river administrators of MLIT. Later, a disaster response exercise was conducted as the final step of the BECAUSE model, E, to verify the effectiveness of the constructed regional cooperation system [13]. However, wide-area evacuation dealt with at this time was a case in which residents in one municipality evacuated to another municipality nearby.

### 5.2. Final Goal of Risk Communication Based on the BECAUSE Model

The final goal of this research was to implement effective risk communication between stakeholders to unify their intentions and to verify effectiveness, in order to make the Kofu Basin resilient to floods. In this study, in order to formulate a wide-area evacuation plan, the author considered it effective to use a bottom-up method in which residents who evacuate specifically decide on the evacuation action that they should take for wide-area evacuation. That is, if there is a concrete wide-area evacuation plan formulated by the residents themselves, the corresponding wide-area evacuation plan of the municipalities supporting it can be easily considered. This further facilitates the formulation of an effective and comprehensive evacuation plan by the prefectural government. 

With the community disaster management plan established by the Riverside District in Chuo City, a plan for supporting specific wide-area evacuation became possible. The BECAUSE model was implemented to clarify the roles that Chuo City, Yamanashi Prefecture, and MLIT should play in risk communication.

### 5.3. Risk Communication from BE to S

The author conducted regular visits to the mayor and the City Crisis Management Section of Chuo City and provided information on the community disaster management plan of the wide-area evacuation considered in the Riverside District. The author asked Yamanashi Prefecture to visit the Riverside District and understand the activities of its residents toward the development of the community disaster management plan. The Kofu River and National Road Office of MLIT cooperated with the author’s risk communication activities. More importantly, the office coordinated and created an opportunity for the author to provide an awareness of the necessity of wide-area evacuation to the mayors in the Kofu Basin at the “Disaster Mitigation Council in the Fuji River Basin”. As mentioned earlier, risk communication was implemented by sharing information with these stakeholders appropriately and encouraging each stakeholder to prepare in advance (Preparation BEfore risk communication). Throughout all stages of the risk communication, the author tried to maintain a good relationship with each stakeholder, taking into account the positions of the stakeholders (Confidence).

At the “Disaster Mitigation Council in the Fuji River Basin” held on 24 April 2018, the author had an opportunity to explain to the mayors of the Fuji River Basin including that of Chuo City (Figure 8). On this occasion, the author provided the following explanation by showing the inundation area map of the Kofu Basin due to the expected maximum rainfall:

(1) The debris flows from the mountainous rivers into large rivers in the Kofu Basin such as the Kamanashi River and the Fuefuki River. Such an overflow causes flooding with debris flow in the Kofu Basin. (2) The whole area from the center of Kofu City to the south will be inundated, accompanied by a wide distribution of assumed areas where houses would collapse due to the intense water flow on the left and right banks of the Kamanashi River. (3) Once the river overfloods, the flooded water will be stored due to the constriction of the river channel in the south of the junction of the two rivers. Thus, inundation will continue for roughly three days. During this period, residents are prompted to evacuate in a wide area in advance.

In 2013, the author implemented a disaster management exercise on wide-area evacuation caused by floods of planned scale, with the participation of MLIT, Yamanashi Prefecture, Chuo City, Kofu City, Ichikawa–Misato Town, and Kofu District firefighting headquarters. For this exercise, the author explained the clarity of these important issues [13]: (1) There is a shortage of police officers for traffic control. (2) Rescue operations using helicopters are not possible in the case of heavy rain. (3) Sharing information among evacuees and neighboring municipalities ensures the support of these municipalities for those who need help for in evacuation.

Finally, the author explained to the mayors that a community disaster management plan for wide-area evacuation without a single casualty was established by the third Residents’ Association of the Riverside District in Chuo City. They were made aware of the scheme by explaining the necessity of structural measures such as constructing a flood control area in the form of a water reservoir, cutting off the water flow, changing its direction by constructing a road embankment, etc.; however, evacuation by non-structural measures is more important than anything (Awareness).

Prior to a panel discussion of the disaster management symposium described later, the author individually interviewed and exchanged opinions with the mayor of Chuo City, the director of the Disaster Management Bureau, Yamanashi Prefecture, and the director of the Kofu River National Highway Office, MLIT. The most serious issue regarding evacuation is that people do not evacuate even when evacuation information is issued. Furthermore, the author explained that stakeholders, such as local and national governments, should play a role in wide-area evacuation, as summarized in Figure 9, if all the residents in the basin are to implement the scheme (Understanding and Satisfaction).

### 5.4. Enactment

The purpose of risk communication is for district residents, of Chuo City, Yamanashi Prefecture, and MLIT, to declare the role they should play in large-scale flood evacuation in the Kofu Basin, along with recognizing each other’s role. Essentially, the wide-area evacuation scheme should be shared by citizens as much as possible. Apart from the wide-area evacuation scheme, various non-structural measures such as a legal-based system, and structural measures, such as raising residential land and roads, and high-rise buildings, as waterproof measures, may be future targets to improve the resilience of the Kofu Basin toward floods. Therefore, it is important for both panelists and citizens to discuss what should necessarily be done to create a flood-resilient Kofu Basin. The author held a disaster management symposium on 25 December 2018 and planned a panel discussion titled “Make use of lessons learned from Mabi-cho to wide-area evacuation in the Kofu Basin.” which included the chairman of the 3rd Residents’ Association of the Riverside District, the mayor of Chuo City, the director of the Disaster Management Bureau, Yamanashi Prefecture, the director of the Kofu River National Highway Office of the MLIT, and the author as panelists, to discuss the role that each stakeholder should play. The chairman of the 3rd Resident’s Association of the Riverside District stated that the most important issues in wide-area evacuation with no casualties are evacuation support for those who need help during evacuation and the development of a wide-area evacuation plan from the 3rd Residents’ Association for the entire Riverside District. Additionally, the chairman asserted that Chuo City should develop this plan throughout the city in the next stage and further requested Chuo City to issue early evacuation information for citizens to initiate the evacuation plan and secure the evacuation destinations.

The mayor of Chuo City stated the essence of assessing transportation means for evacuation, given the wide-scale plan, and that the city was initiating an interregional evacuation study, in cooperation with the neighboring Showa Town with the support of MLIT, within the framework of the Disaster Mitigation Council in the Fuji River Basin. Further, the mayor expressed an interest in the continuous assessment of the wide-area evacuation plan with the cooperation of Yamanashi Prefecture and MLIT.

The director of the Disaster Prevention Bureau, Yamanashi Prefecture stated that the role of the prefecture is to actively advise and support the development of the evacuation plan of municipalities, establish a forum for consultation between related organizations, and ensure consistency among the municipalities in implementing the evacuation plan. Moreover, the director stressed that it is the prefecture’s responsibility to inform residents of the wide-area evacuation plan and implement an evacuation exercise according to the plan.

The director of the Kofu River National Highway Office stated that the role of the office is to advise Yamanashi Prefecture on the process of examining evacuation criteria for wide-area evacuation. Therefore, criteria that trigger wide-area evacuation are to be decided at this examination stage. In the event of a flood, MLIT should inform that such criteria are reached.

### 5.5. Author Proposals for a Flood-Resilient Kofu Basin 

After successful enforcement of the Disaster Management Basic Regulations of Yamanashi Prefecture in April 2018 [14], the author proposed the enactment of urban promotion regulations promoting a flood-resilient Kofu Basin. For example, according to the basic principles of the regulation, wide-area evacuation should be implemented in the following manner. Firstly, priority should be given to evacuation support for those who need help during evacuation. Secondly, there should be an assurance that all residents in the Kofu Basin will be evacuated to prevent the occurrence of even a single casualty. Thirdly, such a wide-area evacuation plan should be formulated, simulated, and implemented with effective collaboration between both residents and governments. This design should be recognized, accepted, and proposed for implementation by Yamanashi Prefecture. Further, the prefecture should declare the regulations for Kofu Basin to promote its resilience against floods, as established by Shiga Prefecture [15,16].

On the basis of the Disaster Management Basic Regulations of Yamanashi Prefecture [14], the damage brought about by the river overflowing into the Kofu Basin can be minimized by assigning top priority to protecting human life over any other development (such as land modification, etc.), such as Nara Prefecture is implementing based on the ordinance [17]. Roadways should be designed to serve as secondary embankments. In line with the objective of resilience, such an idea was proposed by the author, emphasizing the need to strengthen the embankment, build waterproof facilities, and establish a reservoir from the mountainous area to the basin to temporarily store water in rice fields [18]. Additionally, the prefectural government should enact guidance to appropriate residential areas and make the inundation area waterproof. As most of the Kofu Basin area gets inundated, cooperation between residents and the government is of utmost importance. Finally, green infrastructures instead of gray concrete infrastructures should be built to improve the resilience of the basin against floods, which was promoted with the establishment of the Hospitality Yamanashi Tourism Promotion Ordinance by the Yamanashi Prefecture [19]. In line with this, the gray concept could be substituted with green infrastructures for tourism purposes. The promotion ordinance may be a long step away, but with push and focus, the realization of resilience of the basin against floods is achievable.

### 5.6. Evaluation of Participants’ Level of Understanding

Figure 10 is a screenshot of a TV program in which the author presents the proposal among the panelists during the symposium. The discussion was widely broadcast through local TV stations and news programs. The panel discussion was also geared toward enabling the stakeholders (also participants of the symposium) to understand their roles in wide-area evacuation and flood-resilient urban development. Therefore, a survey was conducted with approximately 100 participants of the symposium to confirm their understanding, with an 82% retrieval rate. Figure 11 shows the sectoral breakdown of the stakeholder respondents.

The question, “Do you think that wide-area evacuation is necessary?”, had 95% respondents answering either “necessary” or “necessary to some extent”, which confirms sufficient participant understanding of the purpose of the panel discussion. Around 47% asserted that community residents embody “Who will be the main body in formulating a regional evacuation plan?”. In agreement, 75% of the residents confirmed being the main body, which confirms the validity of the author’s proposal. Accordingly, the question “Which stakeholder has the most important role in wide-area evacuation?” was answered by stakeholders in the following distribution. In favor of Yamanashi Prefecture, 50% of the respondents were from municipalities, 24% from community residents, and 10% from Yamanashi Prefecture. In favor of municipalities, 48% were from community residents, 24% from Yamanashi Prefecture, and 20% from municipalities. Therefore, even though Yamanashi Prefecture assumes that evacuation measures will basically be taken up by municipalities given a wide-area evacuation, the municipalities, in turn, believe that several measures are to be entrusted to Yamanashi Prefecture. This indicates that further risk communication between the prefectural government of Yamanashi and the municipalities is essential. At the stage of Awareness and Understanding of the BECAUSE model, therefore, it is necessary to add some process to eliminate the conflict between Yamanashi Prefecture and municipalities.

Figure 12 provides a chart illustration to a distribution of answers to the question on what issues need to be solved in order to achieve wide-area evacuation. Most of the respondents stated that residents do not want to evacuate, followed by difficulty in securing public shelters and traffic congestion. The answers of all the respondents and those of only the residents were almost identical. In particular, 49% of the municipal staff responded, “Residents do not evacuate,” which implies that municipalities are usually annoyed because residents do not evacuate even when evacuation information is issued. Respondents from Yamanashi Prefecture pointed out the element of securing public shelters as a reason, at 43%. Thus, securing public shelters may be recognized as a role that Yamanashi Prefecture should undertake.

Figure 13 illustrates the distribution of responses to the question asked in multiple answers as “What is necessary to make the Kofu Basin resilient to flood?” The answers by all respondents were in the order of efforts by community residents (30%), followed by citizens’ consciousness reform for disaster management (26%), structural measures by public organizations (18%), the legal system (13%), and structural measures by individuals (12%). Self-help and mutual help as efforts by community residents and citizens’ consciousness reform for disaster management accounted for 56%, structural measures (both public and individual) roughly 30%, and the legal system 13%. Within Yamanashi Prefecture, the same were perceived for the first two answers, but the third was legal systems, at 17%. Most respondents who prioritized the legal system were from Yamanashi Prefecture accounting for 36%. Thus, such a system can be enforced by establishing an ordinance to move residents to a safe place from a highly inundated area, and to require residents to take waterproofing measures by providing financial support. Yamanashi Prefecture easily understood that it could play the role of establishing such an ordinance, as exemplified by the Ordinance of River Basin Management established by Shiga Prefecture [15,16].

## 6. Comparison between Previous and Present BECAUSE Models

The BECAUSE model was developed for the construction of a regional cooperation system for wide-area evacuation, classifying three groups: the disaster-affected municipality; the neighboring municipalities supporting it; and the supporting organization, comprising the disaster management and construction departments of the prefectural government, prefectural police headquarters, and the river administrators of MLIT. A disaster response exercise was conducted using the information system developed by the author as the final step of the BECAUSE model (E: Enactment). After the exercise, Yamanashi Prefecture introduced a comprehensive disaster information system.

The BECAUSE model developed in this study clarifies the role of each organization that supports the wide-area evacuation behavior of the district residents. Therefore, risk communication was conducted individually for each organization, and each organization was requested to consider the specific issues that would support the regional evacuation planned by the district residents. Finally, as part of the Enactment stage a symposium was held, and related organizations, including district residents, met for the first time and were asked to confirm their roles.

In Japan, municipalities have a responsibility to conduct the operation for evacuation of residents with the support of the prefectures. However, when a disaster spans multiple municipalities, the prefecture is to take on that responsibility, and when a disaster spans multiple prefectures, it is the Japanese government who takes on that role. Furthermore, either the Ministry of Land, Infrastructure, Transport and Tourism or the prefecture takes charge of river administration, and river patrols at the time of a flood are undertaken by the municipalities together with the river administrator. These measures are a result of Japan’s disaster response as enacted within the legal system, such as the Disaster Countermeasures Basic Act, the River Act, and the Water Control Act. However, when large-scale river flooding occurs, there is no doubt that quick response in a system based on a pre-existing wide-area evacuation plan can significantly reduce the damage, when compared to gradually shifting the disaster response system from the municipalities to the prefecture. Thus, it is necessary to establish a wide-area evacuation system by applying the risk communication method prior to the occurrence of large-scale river flooding.

## 7. Conclusions

The author proposed the implementation of a risk communication approach by using the CAUSE model to evacuate community residents and the BECAUSE model to evacuate all residents within the Kofu Basin during a large-scale flood. The main findings of the study are summarized below:(1)In the Riverside District in Chuo City, the author proposed and implemented new processes based on the CAUSE model and implemented these twice for wide-area evacuation plan development. The first process covered the evacuation plan of the 3rd Residents’ Association, and the second process expanded it to the entire Riverside District (1400 households 4000 people). As a result, the community disaster management plan of the whole district on the wide-area evacuation plan was formulated.(2)The author applied the BECAUSE model to implement risk communication to promote wide-area evacuation plan formulation and proposed a new bottom-up process with a specific evacuation plan formulation initially by community residents. In the process, the author suggested the role of each stakeholder, such as residents in Riverside District, Chuo City, Yamanashi Prefecture, and MLIT, for the wide-area evacuation. This suggestion was accepted and mutually confirmed by involving specific sectors in a panel discussion via a disaster management symposium.(3)A questionnaire survey of participants in the symposium indicated that 95% of respondents believe that such wide-area evacuation is “necessary” or “necessary to some extent” where it was confirmed that the purpose of the panel discussion was sufficiently understood by the participants. Moreover, 47% of the respondents stressed that community residents should be the main body in formulating a regional evacuation plan. Around 75% of the resident respondents confirmed that they should be the main body, which validates the proposal of the author. Although Yamanashi Prefecture assumed that evacuation measures are basically the role of the municipalities even during wide-area evacuation, the municipalities believed that several issues should be entrusted to Yamanashi Prefecture given the scale. This suggests that further risk communication between the two is essential to effectively formulate an evacuation plan.(4)The author proposed that both structural and non-structural measures should be implemented steadily if the basic concept of the Kofu Basin being resilient to floods is to be taken into account. This was displayed with the enforcement of the Kofu Basin Promotion Ordinance.

If the wide-area evacuation plan is not specific or realistic, then community residents will lose interest in its implementation, thus making it a “painted rabbit”. On the contrary, residents do get involved in an intrinsic, concrete activity that is supported by municipalities and that involves clear and comprehensive coordination with Yamanashi Prefecture. In this way, Yamanashi Prefecture would be able to formulate a highly effective wide-area evacuation plan. As a result, Yamanashi Prefecture will be able to formulate a highly effective wide-area evacuation plan. In this paper, the consensus building process in such a bottom-up wide-area evacuation planning was demonstrated and its effectiveness was verified.

The author applied Rawan’s proposed CAUSE model to support the formulation of community disaster management plans for residents. The S of Rawan’s CAUSE model is the stage of accepting solutions proposed by government or experts. However, the S in the CAUSE model proposed by the author is not only a stage in which the residents accept proposals from the administration or experts but also the stage where residents themselves propose solutions by drawing on indigenous responses by the district residents. Moreover, the two occasions of risk communication carried out in this study were characterized by a questionnaire survey conducted at the Awareness and Understanding stages of the CAUSE model. Representatives of district residents in charge of community disaster management plan felt uneasy whether many residents in their district agreed with the plan development policy. Therefore, the questionnaire results showing that many district residents agree with their proposed plans have become a great impetus for district representatives to further promote formulating community disaster management plan.

With regard to the BECAUSE model, it is also necessary to revise the content and form of risk communication conducted at each stage in accordance with the final goal of Enactment, and this paper has provided such an example. In the risk communication previously proposed by the author, the stages A, U, and S were applied to organizational groups working in cooperation with each other, but in the risk communication in this study, it carried out individually for each organization.

## Figures and Tables

**Figure 1 ijerph-16-02466-f001:**
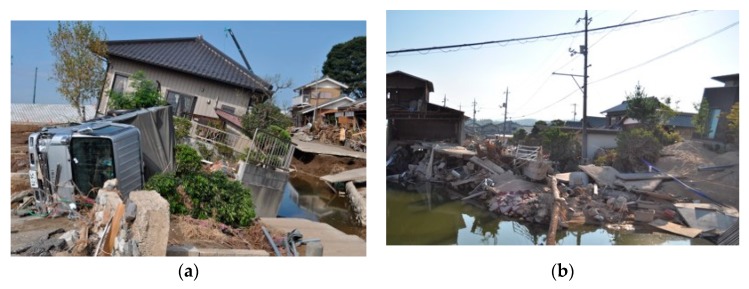
Damage to houses and properties near river banks or break points caused by large-scale flooding. (**a**) Joso City at the height of the 2015 Kanto Tohoku heavy rainfall; (**b**) Mabi-cho, Kurashiki City during the 2018 western Japan heavy rainfall.

**Figure 2 ijerph-16-02466-f002:**
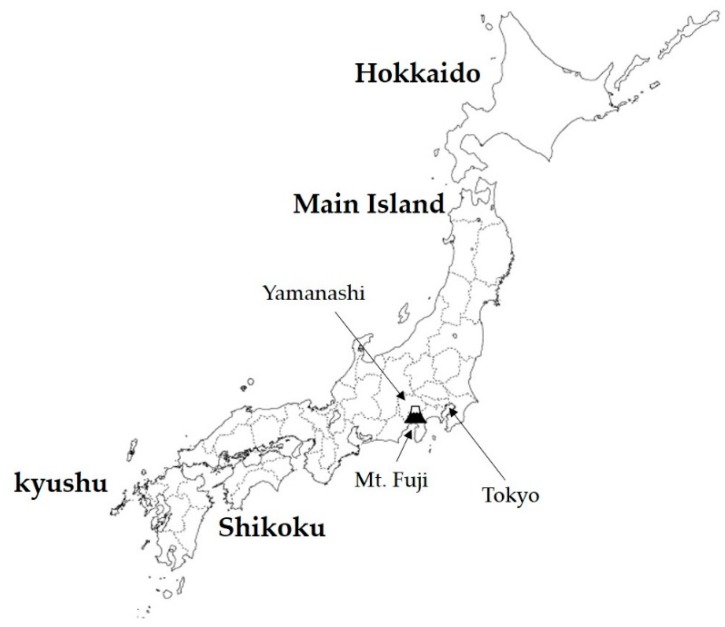
Japanese Archipelago and location of Yamanashi Prefecture.

**Figure 3 ijerph-16-02466-f003:**
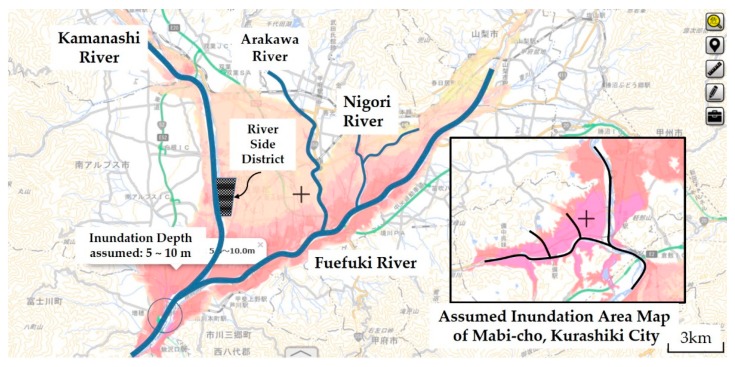
Assumed inundation map of the Kofu Basin compared with that of Mabi-cho.

**Figure 4 ijerph-16-02466-f004:**
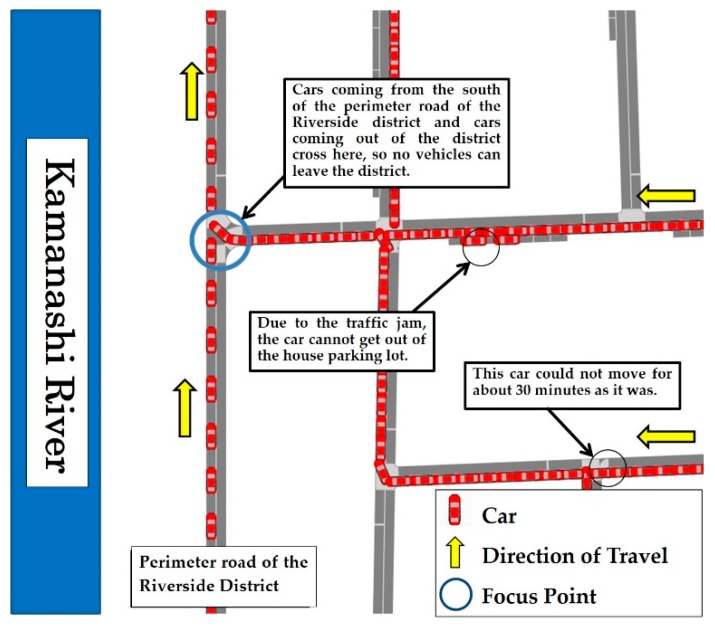
A screenshot of evacuation simulation by car used for explanatory material.

**Figure 5 ijerph-16-02466-f005:**
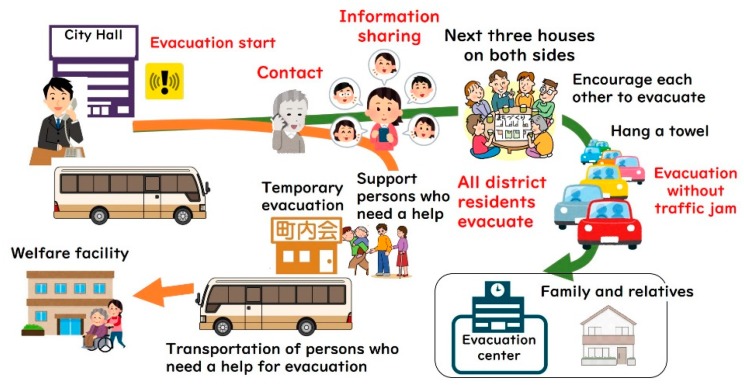
Schematic representation of the wide-area evacuation plan of Riverside District.

**Figure 6 ijerph-16-02466-f006:**
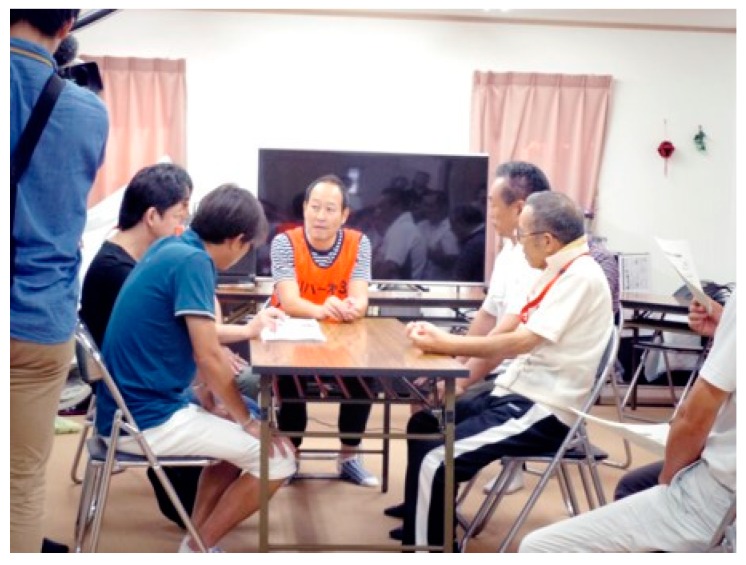
Build-up of a support system for persons who need help in an evacuation.

**Figure 7 ijerph-16-02466-f007:**
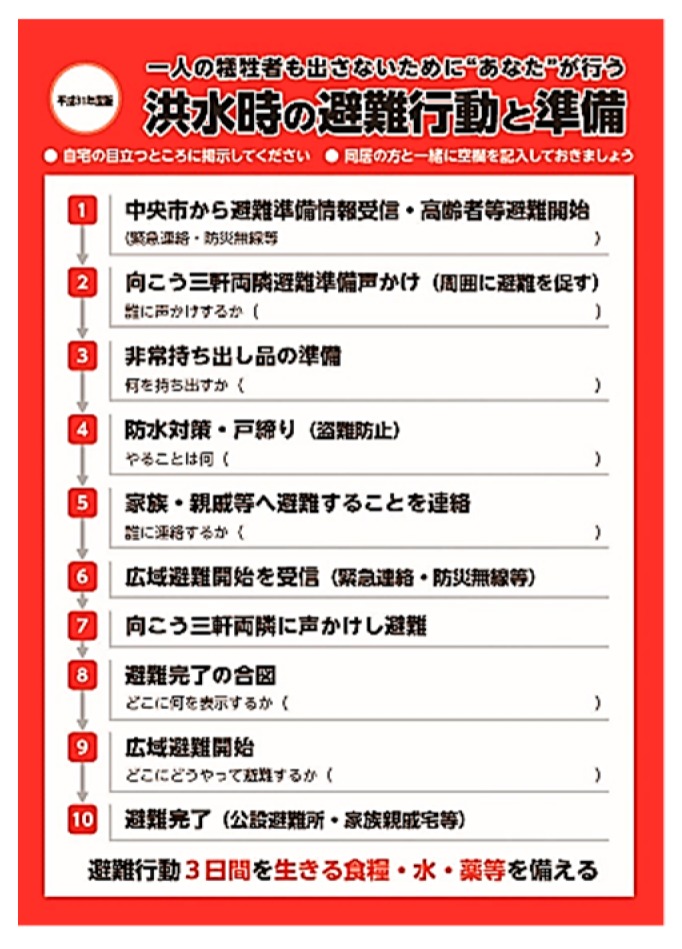
My timeline for each household printed by the residents‘ associations.

**Figure 8 ijerph-16-02466-f008:**
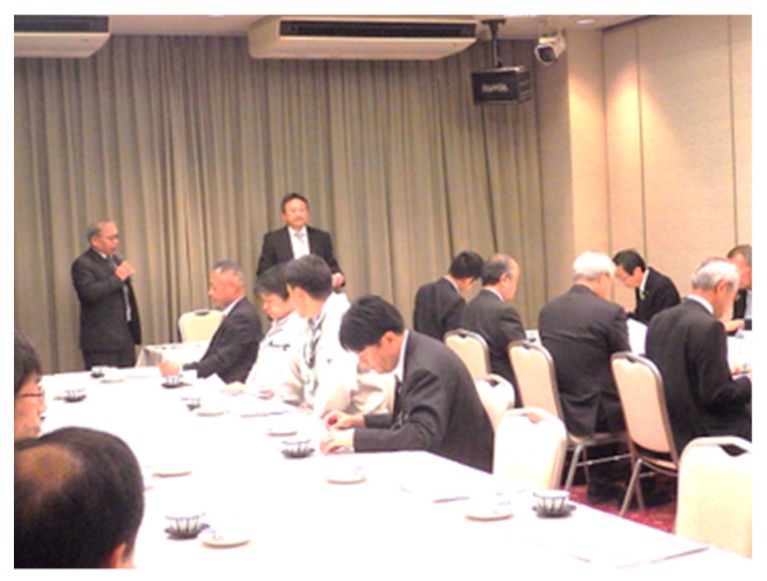
Presentation by the author at the annual meeting of “Disaster Mitigation Council in the Fuji River Basin” held on 24 April 24 2018.

**Figure 9 ijerph-16-02466-f009:**
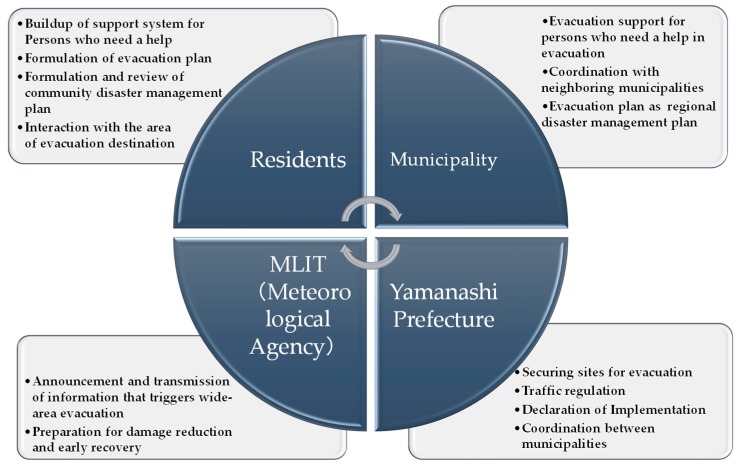
Roles that stakeholders, such as local and national governments, should play in the wide-area evacuation plan proposed by the author.

**Figure 10 ijerph-16-02466-f010:**
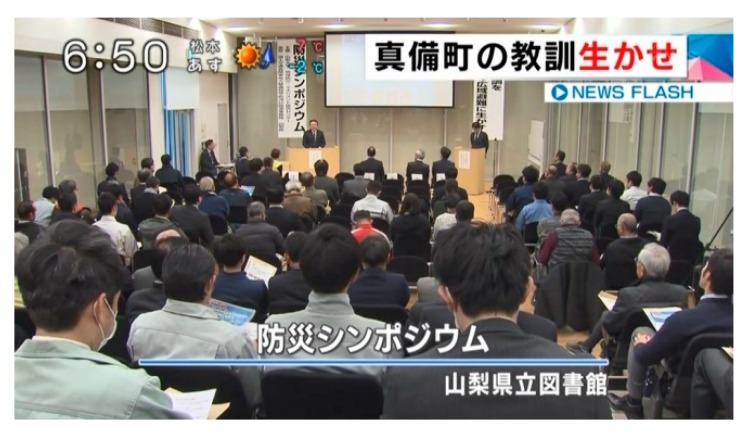
Disaster management symposium.

**Figure 11 ijerph-16-02466-f011:**
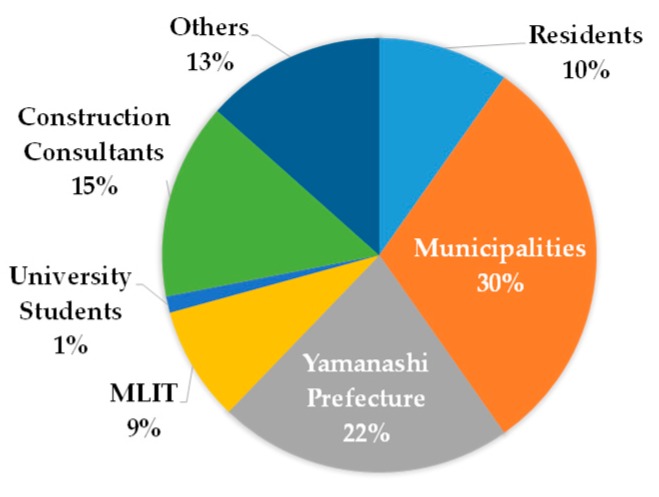
Sectoral breakdown of respondents.

**Figure 12 ijerph-16-02466-f012:**
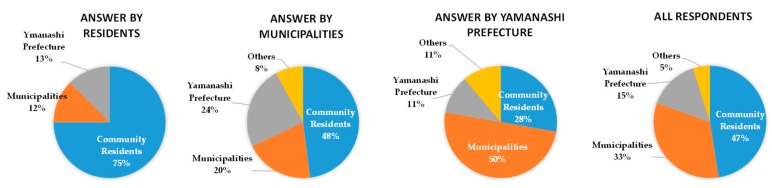
Distribution of responses by sectors on “Which stakeholder has the most important role in wide-area evacuation?”.

**Figure 13 ijerph-16-02466-f013:**
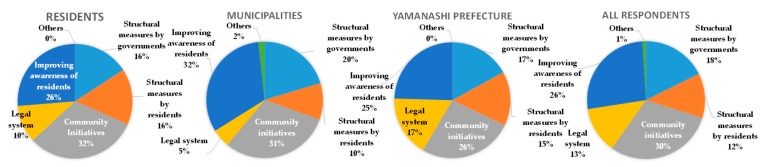
Answers to the question “What is necessary to make the Kofu Basin resilient to flood? (multiple answers allowed)”.

**Table 1 ijerph-16-02466-t001:** Transition of demographics in the Riverside District.

	Year	1978	1986	2002	2005	2019
Association						
1st		162	425	591	577	(600)
2nd			614	863	894	(1000)
3rd			519	1624	1717	(2400)

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
