# Peer review of "Facilitating Community Risk Communication for Wide-Area Evacuation during Large-Scale Floods"

_ijerph, 2019, doi:10.3390/ijerph16142466_

Round 1

Reviewer 1 Report

General Comments

- Perhaps better labelled as an essay or other type of publication rather than an article as it is more about application than study

- Appropriate content that is relevant; could be more aligned with the body of literature and offer connections to issues that are shared globally

- English delivery needs some work and clarification; references need to be expanded to adequately address the aspects covered in the manuscript

- Paper has worth and can further the literature; needs to be tightened while connecting to broader issues/implications of the subject matters

Specific Comments

- Paper is primarily about risk communication, suggest removal of "Disaster Resilience through" from the title to be accurate

- page 1, abstract: English clarity issues

- page 1, lines 20-21: "this study describes a proposed risk communication method implemented" is awkward; this study applies several methods to create?

- page 1, lines 24-26: unclear, needs to be rewritten/clarified

- page 1, line 42: reference to support this statement would be helpful

- page 2, lines 1-2: "has appeared to increase" is insufficient; need references with regard to "global weather anomaly"

- page 2, lines 6-8: it is much more complex than this; see recent MDPI literature with regard to hazards, responses, and related topics to give a better discussion with appropriate sets of references

- page 2, line 14: reference? line 15: how so? line 18: references?

- page 2, lines 9-15: this is an opportunity to address policy development and evolution to be more comprehensive and to be supportive of lines 20-25; see MDPI literature

- page 2, lines 24-25: why "nonetheless require..."? also, does the warning system not exist? if so, why not? (as you've implied Japan has a leading position previously)

- page 2, lines 31-36: there are references that could be cited here to support how this is a commonality shared among a global population

- page 2, line 40: meaning 42 of the elderly you've referred to?

- page 3, lines 8-9: use references to broaden this thought

- page 3, lines 10-13: references needed

- page 4, line 21: how does that rate compare with rates not due to disasters? to rates nationally? to rates in other countries?

- page 5: can you provide some population demographics of the areas/people surveyed?

- page 5, line 30: why not include as appendix material?

- page 6, lines 3-8: some would argue as to the roles of television and radio; others with regard to social media's roles; what about in your areas?

- page 6, line 26: show it?

- page 8, line 6: rather than "research" this is more application or prototype or beta-test with implementation?

- page 8, line 8: is it the "Kofu Basin resilient to floods" or the population? Or the socio-economic system found within the basin?

- page 8, section 5.3: however preparation before is not equivalent to understanding; risk communication is meant to drive responses

- page 8, line 30: more than "Confidence" is the self-determined significance of a threat and thus need for a response; there is much literature with regard to social responses to risk communications, their effectiveness, and salient issues; some consideration of these must be included for proper context and expectations

- page 9, lines 28-29: see prior comment above; cannot leave this without comment and context

- page 10, line 8: same as page 8, line 8 comment

- page 10: can this material be summarized in a timeline format? or perhaps tabular with specific milestones in the process relative to deliverables and outcomes? otherwise this reads as more of a diary-like narrative or personal accounting which detracts from the paper's purpose

- page 10, section 5.5: was there a section 5.4?

- page 10, lines 38-42: again this speaks to policy implications (and as per page 2, lines 9-15 comment)

- page 11, lines 1-12: other than the policy implications, this seems beyond the scope of this paper as it is 'next steps' and not truly addressed earlier in the paper; omit?

- page 11, line 13: or is this really "5.6. Developing Resilience"? The paper loses focus on this page

- page 12, lines 20-33: again policy, in this instance development of policy; see recent MDPI papers as to policy processes

- page 13: this reads more as a summary rather than drawing specific conclusions or making of recommendation; please rewrite

Author Response

According to the comment by the reviewer, I removed “Disaster Resilience” from the title of article.

I think that there are many reasons for the lack of expressive ability in my English as a reason why the referee's understanding was not obtained. However, there are still some areas in which the reviewers' comments on policy may not be accepted. As measures against flood damage, what I introduced on the article, and the information provided to the mayor and Yamanashi prefecture, is that the Ministry of Land, Infrastructure, Transport and Tourism, Shiga Prefecture, Nara Prefecture, etc. are already working on it. The suggestions I have made for Chuo City and Yamanashi prefectures are by no means policy recommendations.

Such comments are given at the end paragraph in the new chapter 6.

At the stage of English proofreading, there was arbitrary translation. It was clearly rewritten with the wrong content. I missed to check it and I apologize to referee for making troubles. So, Abstract and has completely replaced my original texts and all the sentences that the referee thought to be in the discussion were replaced with the original ones.

Comment regarding Page4 line 21:

I rewrote the sentences according the comment.

Comment regarding Page5: population demographics

Now, I am asking a survey to investigate, but accurate statistics are not available.

Page 6 comment on roles of media

I added descriptions at the end of Chapter 4.

Request to show simulation in Page 6

I added an example of evacuation simulation by car as Figure 4.

Comments on page 8:

I confirmed that my paper is suitable for an article and the editor of the journal judged that my paper is suitable for Demonstrated Community Resilience based on the abstract that I submitted in advance.

Comment regarding page 13 on concluding remarks:

I added two paragraphs as conclusion.

Reviewer 2 Report

The manuscript provides a good example about the survey in the field of disaster evacuation and management. However, the following comments could be explained in the manuscript to improve the academic contribution.

-      Please describe how the authors improve BECAUSE model to fit in the study basin. Areal specific characteristics should be explained when BECAUSE model was applied.

-      The survey results was shown in the Figures. Please describe how the results were implemented in the BECAUSE model.

-      The survey itself is very good practice, but there should be some points to improve the wise evacuation process using the survey results.

Author Response

Dear Reviewer:

I appreciate your comments on my article.

Comment regarding the improvement of the BECAUSE model

→A new chapter 6 has been inserted to explain the comparison with previous BECAUSE models.

Comment on the reflection on the modification of the BECAUSE model.

→The survey results were conducted to confirm the effects of model application. So, application of the survey result to the implementation of the BECAUSE model is a future task.

Comment on the improvement of the model.

→I added a comment on the improvement in Chapter 5.6, which is highlighted.

Round 2

Reviewer 1 Report

Appreciate the changes and additions made by the authors as they have made for a stronger paper with more clarity.

There are still a number of instances where additional references should be added and the discussion can link the work completed to the 'bigger picture' and to policy (both of relevance to the reading audience). Suggest re-reading of first round comments by page/line for direction as to where additional literature can be cited.

The English language improvements have helped; suggest that the remainder be part of the editorial process to ensure smooth reading.

The authors have made important and useful clarifications that have made a difference in the presentation of the work.

Thank you for the strong efforts towards improvement...

Author Response

I added a few references.

I also added Table 1 showing the transition of demographics in the Riverside District

I revised my paper on the corrected sentences, asking a professional English proofreading firm (highlighted and a certificate attached).